# ALK Inhibitors-Induced M Phase Delay Contributes to the Suppression of Cell Proliferation

**DOI:** 10.3390/cancers12041054

**Published:** 2020-04-24

**Authors:** Sirajam Munira, Ryuzaburo Yuki, Youhei Saito, Yuji Nakayama

**Affiliations:** Department of Biochemistry & Molecular Biology, Kyoto Pharmaceutical University, Kyoto 607-8414, Japan; kd16015@ms.kyoto-phu.ac.jp (S.M.); yuki2019@mb.kyoto-phu.ac.jp (R.Y.); ysaito@mb.kyoto-phu.ac.jp (Y.S.)

**Keywords:** ALK, M phase, crizotinib, SAC, misalignment, cell proliferation, RO-3306

## Abstract

Anaplastic lymphoma kinase (ALK), a receptor-type tyrosine kinase, is involved in the pathogenesis of several cancers. ALK has been targeted with small molecule inhibitors for the treatment of different cancers, but absolute success remains elusive. In the present study, the effects of ALK inhibitors on M phase progression were evaluated. Crizotinib, ceritinib, and TAE684 suppressed proliferation of neuroblastoma SH-SY5Y cells in a concentration-dependent manner. At approximate IC_50_ concentrations, these inhibitors caused misorientation of spindles, misalignment of chromosomes and reduction in autophosphorylation. Similarly, knockdown of ALK caused M phase delay, which was rescued by re-expression of ALK. Time-lapse imaging revealed that anaphase onset was delayed. The monopolar spindle 1 (MPS1) inhibitor, AZ3146, and MAD2 knockdown led to a release from inhibitor-induced M phase delay, suggesting that spindle assembly checkpoint may be activated in ALK-inhibited cells. H2228 human lung carcinoma cells that express EML4-ALK fusion showed M phase delay in the presence of TAE684 at about IC_50_ concentrations. These results suggest that ALK plays a role in M phase regulation and ALK inhibition may contribute to the suppression of cell proliferation in ALK-expressing cancer cells.

## 1. Introduction

Human anaplastic lymphoma kinase (ALK) is a receptor-type tyrosine kinase (RTK) composed of 1620 amino acids and belongs to the insulin receptor superfamily. The expression of ALK is especially high in neonates and is found in discrete areas of the developing central and peripheral nervous systems [1]. ALK expression is also reported in the adult mammalian hippocampus [2]. Consistent with this expression profile, ALK is reported to play a regulatory role in the function of the nervous system and in mammalian behavior [3]. Some proteins have been reported as ligands of ALK, including pleiotrophin, midkine, augmentor-β, and augmentor-α [4,5,6]. Like other RTKs, ALK is activated through autophosphorylation after ligand binding and receptor dimerization. The direct biological roles of ALK are not completely understood [7]. 

Genetic alteration of ALK, including gene amplification, gene fusion, and mutation with gain of function, has been identified in different cancers. Activating mutations at R1275, F1174, and F1245 positions, and gene amplification are reported in pediatric cancer neuroblastoma [8,9]. *ALK*-*NPM1* fusion was first identified in anaplastic large-cell lymphoma (ALCL) cell line [10]. Although ALK is not expressed in the normal lymphoid cells, the vast majority of pediatric ALCL patients are ALK-positive [10,11]. Gene fusion is found widely in inflammatory myofibroblastic tumors, diffuse large B-cell lymphoma, and esophageal squamous cell, colorectal, breast, and non-small cell lung carcinomas (NSCLC) [11]. Genetically altered ALKs are commonly activated by dimerization of their fusion partners or by activating mutations in full-length ALK. Activated ALK triggers PI3K-AKT [12], CRKL-C3G [13], JAK-STAT [14], and MAPK pathways [15] in a manner that is dependent on ALK subcellular location and protein stability. Activating mutations in ALK are correlated with greater cell proliferation, resistance to apoptosis, and enhanced DNA synthesis [9,16], which contribute to oncogenesis. Inhibition of ALK with small molecule inhibitors suppresses cell growth of ALK-positive cancers. The US Food and Drug Administration has approved many ALK inhibitors, such as crizotinib, ceritinib, and alectinib, for the treatment of patients with non-small-cell lung carcinoma (NSCLC) [11]. Unfortunately, an acceptable success rate has not been achieved because diverse genetic alterations in ALK affect inhibitor efficacy [17,18]. Furthermore, treatment with crizotinib and ceritinib causes focal amplification of *ALK* [19] and *KRAS* [15] genes, and have also been reported to confer resistance against ALK inhibitors in some cases. This prompted us to investigate the possibility of involvement of chromosome segregation errors in acquired resistance to ALK inhibitors, which remains largely unexplored.

Cell division involves the division of one cell into two genetically identical daughter cells. Duplicated sister chromatids are condensed, aligned at the cell equator and segregated into two cells by an elaborate process involving cytoskeletons, motor proteins, and kinases. To ensure an accurate genetic transmission, an array of serine/threonine kinases, such as cyclin-dependent kinase 1 (CDK1) [20], polo-like kinase 1 (PLK1) [21], and Aurora kinases [22], are involved in several critical steps. To date, however, the involvement of RTKs in M phase regulation has not been broadly investigated. Our current search for compounds that affect the M phase found that crizotinib caused M phase delay. Therefore, to examine whether chromosome segregation errors during crizotinib treatment confer crizotinib resistance to the cell, we studied here the effects of ALK inhibitors (including crizotinib) on cell division. At approximate IC_50_ concentrations, ALK inhibitors delayed M phase progression in prophase/prometaphase and increased chromosome misalignment. Furthermore, spindle assembly checkpoint (SAC) is activated indirectly in ALK inhibitor-treated cells. The observations, thus, suggest that ALK is a new candidate for M phase regulation. ALK inhibitor-induced M phase delay may be partly responsible for the suppression of cell proliferation in cells treated with ALK inhibitors.

## 2. Results

### 2.1. ALK Inhibitors Delay M Phase Progression

One target of crizotinib is ALK tyrosine kinase, the expression of which depends on cell type. To examine the effects of crizotinib on M phase progression, ALK expression was measured in breast cancer MCF7, neuroblastoma SH-SY5Y and IMR-32, and cervical adenocarcinoma HeLa S3 cells. Western blot (WB) analysis with anti-ALK antibody showed several bands (Figure 1A). The molecular weight of full-length ALK is 180 kDa and increases to 220 kDa after glycosylation. The 220 kDa band was observed in SH-SY5Y and IMR-32 cells, but was absent in ALK-negative MCF7 cells [5]. Although some studies have reported that HeLa S3 cells express low levels of ALK [5], we could not detect the full length 220 kDa ALK band in our experiments. Another band at around 140 kDa represents truncated ALK forms produced by extracellular cleavage [23]. However, the physiological significance and the underlying molecular mechanism of the cleavage event are unclear. Truncated ALK is frequently found in neuroblastoma cancer cell lines [8]. We did observe the band corresponding to the truncated ALK protein in SH-SY5Y and IMR-32 cells (Appendix A) with anti-phospho-ALK antibody. Three bands detected in the ALK-negative MCF7 cells [5] were considered as non-specific bands, as a mixture of three siALKs was unable to reduce them (Appendix A). Since the overall expression of ALK is higher in SH-SY5Y than in the other cells, SH-SY5Y cells were used in further experiments.

The effect of ALK inhibitors on cell proliferation was subsequently addressed to determine appropriate inhibitor concentrations for analyzing effects on M phase progression. SH-SY5Y cells were cultured in the presence of ALK inhibitors crizotinib, ceritinib, and TAE684. Cell numbers were estimated using WST-8. Absorbance of reduced 2-(2-methoxy-4-nitrophenyl)-3-(4-nitrophenyl)-5-(2,4-isulfophenyl)-2*H*-tetrazolium monosodium salt at 450 nm showed that inhibitors reduced cell numbers in a concentration-dependent manner. Inhibitor IC_50_s were determined as shown in Figure 1B. 

SH-SY5Y cells were next synchronized with the CDK1 inhibitor RO-3306. Using this method, approximately 30% of cells can synchronously enter M phase. Fixation at different times after release allows visualization of M phase progression [24]. Treatment with RO-3306 continued for 20 h followed by release for 1 h in medium containing either dimethyl sulfoxide (DMSO), 1 µM crizotinib, 0.5 µM ceritinib, or 0.3 µM TAE684. Inhibitor concentrations were approximately equal to their respective IC_50_s. Fixed cells were stained for α-tubulin and DNA, and M phase cells were classified into four groups based on microtubule and chromosome morphology: prophase/prometaphase (P/PM), metaphase (M), anaphase/telophase (A/T), and cytokinesis (Cyto). Most control cells progressed to cytokinesis, whereas inhibitor-treated cells were significantly delayed in prophase/prometaphase (Figure 1C). Mitotic index, the percentage of M phase cells, showed no significant change in inhibitor-treated cells, suggesting that ALK inhibition delays M phase progression without affecting mitotic entry. It is of note that the number of cells with misaligned chromosomes significantly increased during inhibitor treatment (Figure 1D), suggesting that ALK could be involved in chromosome alignment. To confirm that ALK was actually inhibited, autophosphorylation of ALK at Tyr1507 was examined after 1 h treatment with inhibitors. WB analysis showed reduced phosphorylation, confirming the inhibition of ALK kinase activity (Figure 1E). Taken together, these results suggest that crizotinib, ceritinib, and TAE684 delay M phase progression by inactivating ALK, and this activity may be partly responsible for suppression of cell proliferation.

To identify the prolonged sub-phases in ALK-inhibited cells, time-lapse imaging was performed. After release from the RO-3306, cells were continuously observed using an Operetta imaging system in the presence of Hoechst 33342 to monitor chromosome movement. Representative control and crizotinib-treated cells are shown in Figure 2 (also see Appendix A). The M phase was divided into three stages: the prophase/prometaphase (P/PM), metaphase (M), and anaphase/telophase (A/T). The durations of each of these stages are shown in the graph. In the control cells, the average durations of P/PM and M were 31 and 26 min, respectively. In the crizotinib-treated cells, the average durations of P/PM and M were 74 and 83 min, respectively, suggesting that crizotinib treatment prolongs the duration of both P/PM and M. Interestingly, the cells exhibiting abnormal M phase characteristics, including misoriented spindle (23/39 cells) and misaligned chromosomes (20/39 cells), were frequently observed upon crizotinib treatment, suggesting that crizotinib causes M-phase delay by spindle misorientation and chromosome misalignment.

Crizotinib also targets another receptor tyrosine kinase, c-Met, also known as hepatocyte growth factor receptor. To identify the target of crizotinib, c-Met expression in indicated cell lines was examined (Appendix A). c-Met is expressed in HeLa S3 cells but not in SH-SY5Y cells, suggesting that the target of crizotinib may be ALK in SH-SY5Y cells. Crizotinib caused a significant mitotic delay in prophase/prometaphase in HeLa S3 cells (Appendix A). However, c-Met knockdown using three different siRNAs targeting c-Met did not show consistent results; only one siMet RNA, but not the other two, caused M phase delay (Appendix A). No conclusion on the involvement of c-Met in M phase regulation could be reached. 

### 2.2. Knockdown of ALK Causes M Phase Delay

In general, inhibitors can cause off-target effects. To identify the target of ALK inhibitors in SH-SY5Y cells, cells were treated with three different siALKs (siALK #1, #2, and #3). WB analysis showed partial knockdown of ALK (Figure 3A). After siRNA treatment for 52 h, cells were synchronized with RO-3306. After a 90 min release, M phase cells were classified into two groups: before (P/PM/M) or after (A/T/Cyto) anaphase onset, based on microtubule and chromosome morphology. Knockdown of ALK increased the number of cells before anaphase onset (Figure 3B, siALK #1, #2, Figure 3C, siALK #3), indicating a delay in M phase. Thus, ALK may be involved in M phase progression. In addition, mitotic index was reduced by ALK knockdown (Figure 2B), which is in agreement with a previous study describing the prolongation of cell cycle progression in G1 phase [25]. Furthermore, stable cells that express HA-tagged wild-type ALK (ALK-HA) upon Dox treatment were established from SH-SY5Y cells (SH-SY5Y/ALK-HA) (Appendix A). Inducible expression of ALK-HA was detected by WB analysis after knockdown of endogenous ALK by 3’-UTR-targeting siALK (siALK #3) in the presence of Dox (Figure 3D). Seventy percent of control knockdown cells (siCtrl) began chromosome segregation, i.e., they reached anaphase or later stages. Knockdown of endogenous ALK increased the number of cells that had not progressed to or through anaphase (P/PM/M). Re-expression of ALK-HA rescued ALK knockdown-induced delay (Figure 3C), confirming that ALK is involved in M phase regulation. Mitotic index was not rescued by ALK re-expression. Possibly, another target of ALK may be involved in regulating mitotic entry, which could not be rescued by moderate ALK re-expression.

### 2.3. Depletion of ALK Prolonged both Prophase/Prometaphase and Metaphase

To investigate the underlying mechanism of ALK knockdown-induced M-phase delay, time-lapse imaging was performed with SH-SY5Y cells first treated with siALK #1 and nontargeting siCtrl and then synchronized with RO-3306. Representative images of control and ALK knockdown cells are presented in Figure 4A. The M phase was divided into three stages as described in Figure 2, and the durations of each stage and percentages of cells in each stage were plotted (Figure 4B). In control cells, average durations of P/PM and M were 29 and 23 min, respectively. In ALK-knockdown cells, P/PM was prolonged to 63 min, more than twice the time observed for control cells. The duration of metaphase in ALK-knockdown cells was more than 40 min (average value), which was greater than that observed in control cells. The line graph of the results of time-lapse imaging showed that the peak of M decreased and those of P/PM and M did not decline to zero, suggesting prolongation of P/PM and M. Further, five among 31 ALK-knockdown cells failed to complete M phase (black dashed line). These results suggest that ALK knockdown prolonged the duration of both prophase/prometaphase and metaphase.

### 2.4. SAC Is Activated Indirectly in ALK-Inhibited Cells

SAC is activated in cells with unaligned chromosomes to promote the proper segregation of chromosomes [26]. Inhibition of ALK delayed M phase progression and increased the number of cells with misaligned chromosomes. Hence, M phase delay could be accompanied by SAC activation. To investigate, cells were treated with crizotinib alone or in combination with the MPS1 inhibitor, AZ3146 [27]. Cells were synchronized with RO-3306 and treated with crizotinib with or without AZ3146 during a 45 min release period (Figure 5A). Less than 30% of solvent control cells (DMSO), but most AZ3146-treated cells progressed to cytokinesis, suggesting that SAC inactivation causes immature mitotic exit. Crizotinib-treated cells showed delayed P/PM compared with control; however, many crizotinib-treated cells progressed to cytokinesis under simultaneous AZ3146 treatment (Figure 5A). This inference was supported by knockdown experiments of MAD2, which is a component of the mitotic checkpoint complex (MCC) and is necessary for SAC activation [28]. When cells were transfected with siMAD2, most MAD2 knockdown cells progressed to cytokinesis (Figure 5B, siMAD2) faster than the control cells treated with nontargeting siRNA (siCtrl), suggesting that SAC was inactivated in them. In this experiment, MAD2 knockdown cells progressed to cytokinesis, even when cells were treated with crizotinib. These findings suggest that SAC is activated indirectly in ALK inhibitor-treated cells and participates in the delay of onset of anaphase.

### 2.5. Inhibition of EML4-ALK Fusion Delays M Phase Progression

EML4-ALK is an oncogenic fusion protein involved in the pathogenesis of lung cancer [29,30]. It regulates common signaling pathways with NPM1-ALK, such as ERK/MAPK, PI3K/Akt, and JAK/STAT, resulting in the acceleration of cell proliferation and resistance to apoptosis [29,30,31]. However, the role of EML4-ALK fusion in the regulation of M phase progression has not been investigated. First, H2228 cells were treated with indicated inhibitors, and cell numbers were estimated using WST-8 to determine IC_50_s as shown in Figure 6A. After inhibitor treatment at IC_50_ concentrations, WB analysis confirmed that phosphorylation of EML4-ALK is eliminated with almost no concomitant change in the total expression level of EML4-ALK (Figure 6B).

Time-lapse imaging was performed in asynchronous cells since H2228 cells could not be synchronized with RO-3306. Cells were treated with TAE684 and DMSO for 12 h during imaging, and representative cell images are shown in Figure 6C. M phase cells were classified into three groups, as described in Figure 2 (Figure 6D). DMSO-treated cells were in P/PM for an average of 30 min, and P/PM in TAE684-treated cells was prolonged to an average duration of 130 min. Moreover, only 40% of TAE-treated cells completed M phase irrespective of the prolonged P/PM phase. The remaining cells either failed to complete M phase (37%), died during mitosis (13%), or prematurely exited (10%). These results indicate that EML4-ALK may also be involved in M phase regulation and suggest that targeting an ALK fusion protein suppresses cell proliferation partly via M phase disruption. 

## 3. Discussion

ALK plays an important role in tumorigenesis, and it is expressed at high levels in some types of tumor. In particular, signaling mediated by ALK is essential for cancer cell proliferation and survival [32,33,34,35]. Conversely, normal cells express no or low levels of ALK when ALK is expressed [10,36]. ALK is thus an attractive target for anticancer therapy. ALK inhibitors are used in targeted therapy of ALK-positive cancers, especially lung cancer. Encouraging therapeutic outcomes have been observed in several clinical studies [11]. These inhibitors suppress cell growth by inhibiting signaling pathways downstream of ALK [32,33,34,35]. Our study shows that three ALK inhibitors suppress proliferation of SH-SY5Y and H2228 cells (Figure 1B and Figure 6A). Notably, these inhibitors mostly abolished autophosphorylation of ALK at IC_50_ concentrations in both cell types, implying that ALK activity is essential for proliferation of SH-SY5Y and H2228 cells. 

H3122 and H2228 cell lines are non-small-cell lung cancer cell lines expressing EML4-ALK fusion protein, which are widely used in preclinical cancer research. Compared to H3122 cells, the higher concentration of ALK inhibitors should be used to suppress cell growth in H2228 cell line [37]. As ALK inhibitors suppress ERK phosphorylation in H3122 but not in H2228 cell line, and combination of ALK inhibitor with Met inhibitor suppresses ERK phosphorylation and cell proliferation in H2228 cell line, MEK/ERK pathway may be independent of ALK and responsible for cell proliferation in H2228 cell line [37,38]. Indeed, polytherapy treatment with ALK inhibitor and MEK inhibitor suppresses cell proliferation in H2228 cell line. However, incomplete suppression of cell proliferation may result in resistance to this chemotherapy. Therefore, the development of novel approaches is vital to combat the disease. In the present study, we used H2228 cells to identify a novel mechanism for the suppression of cell proliferation, which may be exploited for developing new approaches of polytherapy. 

This study assessed the effect of ALK inhibition on M phase progression. Small molecule inhibitors caused M phase delay via suppression of autophosphorylation of full-length ALK and EML4-ALK fusion in SH-SY5Y cells and H2228 cells, respectively. Inhibitors often have off-target effects. Crizotinib is known to inhibit c-Met and ROS1 tyrosine kinases in addition to ALK [7]. Ceritinib also targets IGF-1R and ROS1 [7]. In contrast, no off-target effects of TAE684 have been reported. Although complete knockdown was not achieved, treatment with ALK-targeting siRNAs consistently showed M phase delay, similar to the delay associated with inhibitor treatment. Additionally, ALK re-expression rescued siALK-mediated M phase delay (Figure 3C). These results indicate that ALK has a regulatory role in M phase progression. Autophosphorylation in full-length ALK and EML4-ALK was observed (Figure 1E and Figure 6B) and was inhibited by ALK inhibitor treatment. Since ALK inhibitors block ALK kinase activity, effects on M phase progression are thought to be caused by this inhibition. That is, the kinase activity of ALK may be required for M phase regulation. In addition to that, ALK inhibitors caused cell death in M phase in H2228 cells. Therefore, ALK inhibitor-induced M phase disruption may contribute to the suppression of cell proliferation and cell death. 

Knockdown of MAD2 and the Mps1 inhibitor, AZ3146, reversed crizotinib-induced M phase delay (Figure 5A). Mps1, a SAC-regulating Ser/Thr kinase, supports MCC assembly through phosphorylation of Knl1 and therefore plays an essential role in creating docking sites for SAC proteins [28]. Elimination of ALK inhibition-caused M phase delay by MAD2 knockdown and Mps1 inhibitor implies that SAC is activated in ALK-inhibited cells. This checkpoint is activated when spindle formation is defective, for example, when chromosomes are unaligned at the metaphase plate and in the presence of defects in astral microtubule formation and its tethering to the cell cortex [39,40]. In these situations, via inhibition of the APC/C, SAC ensures the binding of microtubules to the kinetochore before anaphase onset by preventing premature mitotic exit [28]. Chromosome misalignment occurs because of defects in spindle formation, such as failure of chromosome congression and defects in binding of microtubules to the kinetochore [41,42]. Time-lapse imaging showed that chromosome alignment at the cell equator took longer or failed in ALK-knockdown cells (Figure 4B). Thus, SAC may be indirectly activated via ALK inhibition-caused defects in spindle formation.

In addition to the M phase delay, a decrease in mitotic index was observed in ALK-knockdown cells. Suppression of mitotic entry, acceleration of mitotic exit, and prolongation of cell cycle can decrease the mitotic index. Inhibitor treatment during release from RO-3306 treatment did not reduce the mitotic index, thereby eliminating direct suppression of mitotic entry and acceleration of mitotic exit as an explanation. It has been reported that ALK inhibitors induce cell cycle arrest in the G1 phase by reducing the gene expression levels of cell cycle-related proteins (Cyclin D1, D3, and E2F1) and increasing gene expression levels of p27 [25]. Therefore, if ALK knockdown prolongs cell cycle length, it results in relatively lower mitotic index. Alternatively, if knockdown affects the expression of proteins involved in mitotic entry at transcriptional and translational levels, knockdown can affect mitotic entry indirectly. Since 3 days were needed to achieve ALK knockdown, M phase-regulating proteins could have been dysregulated at transcriptional and translational levels in ALK-knockdown cells. Further studies will be required to confirm this hypothesis.

Proliferation of cancer cells expressing EML4-ALK fusion typically depends on the downstream signaling of this oncogene, making it an attractive target for chemotherapy treatments. However, durable responses from the cells are uncommon and cells usually acquire resistance to ALK inhibitors via multiple mechanisms, such as mutations in the ALK kinase domain [7,11,43], amplification of *ALK* gene [19] and *KRAS* gene [15], activation of kinases including EGFR [44], IGF1R [45], KIT [46] and Src [47], and downregulation of DUSP6 [15]. All these alterations maintain the activation of ERK and promote cancerous cell growth in the presence of ALK inhibitors. These observations are in agreement with a previous study reporting that SHP2 inhibition restores sensitivity to ALK inhibitors in resistant cancers by inactivating ERK activity [48]. Thus, polytherapy with ALK inhibitor and MEK/ERK inhibitor suppresses cell growth in resistant cancers. However, focal amplification of genes such as *ALK* and *KRAS* raises the question as to whether anti-ALK therapy induces chromosome instability.

ALK inhibition caused defects in chromosome alignment and thereby delayed M phase progression. Defects in chromosome alignment can induce chromosome mis-segregation [49]. Higher levels of chromosome mis-segregation will lead to cell death and suppress cancer cell growth when chromosome instability rises above a threshold level [50,51]. However, chromosome instability below the threshold causes chromosome rearrangement and genetic diversification without causing cell death. Among cells with broad genetic diversity, clones can evolve via acquiring the growth capacity under ALK-inhibited condition. As reported in a previous study, continuous treatment with crizotinib or ceritinib leads to the amplification of the wild-type *KRAS* gene, further supporting this argument [15]. Biopsy from patients possessing acquired resistance to ALK-inhibitor therapy revealed focal amplification of *KRAS* (three of 15 patients). Given that ALK inhibition causes aberrant chromosome segregation, it can accelerate the development of acquired resistance to ALK-inhibitor therapy by producing genetically diverse cancerous cells. ALK inhibitors at approximate IC_50_ concentrations delayed M phase progression. Thus, ALK inhibition-induced suppression of cell proliferation may be partly attributed to M phase delay. Consistent with this notion, combinations of ALK inhibitors with agents that increase chromosome mis-segregation, such as microtubule-stabilizing drugs or inhibitors of PLK1 and Aurora kinases, may improve the efficiency of polytherapy. Thus, further studies are necessary to develop efficient polytherapy treatments. An efficient polytherapy should be able to cause cell death with severe chromosome segregation errors above a threshold level for induction of cell death or, alternatively, should be able to cause cell death without causing chromosome mis-segregation to avoid genetic diversification.

## 4. Materials and Methods

### 4.1. Cells

Human cervical adenocarcinoma HeLa S3 (Japanese Collection of Research Bioresources, Osaka, Japan), Lenti-X 293T (Clontech Laboratories, Mountain View, CA, USA), human neuroblastoma SH-SY5Y (European Collection of Cell Cultures, Salisbury, UK), IMR-32 (Japanese Collection of Research Bioresources), non-small lung cancer H2228 (American Type Culture Collection, Manassas, VA, USA), and breast cancer MCF7 (Japanese Collection of Research Bioresources) cells were cultured in Dulbecco’s modified Eagle’s medium (DMEM) containing 20 mM HEPES-NaOH (pH 7.4) and 5% fetal bovine serum (FBS) at 37 °C in a 5% CO_2_ atmosphere.

### 4.2. Reagents

The inhibitors of ALK tyrosine kinase, crizotinib was obtained from LC Laboratories (Woburn, MA, USA), and ceritinib and TAE684 from Selleck Chemicals (Houston, TX, USA). The Mps1 inhibitor AZ3146 was obtained from Adooq Bioscience (Irvine, CA, USA), and the reversible CDK1 inhibitor RO-3306 from Selleck Chemicals. These inhibitors were dissolved in DMSO (Nacalai Tesque, Kyoto, Japan).

### 4.3. Antibodies

The primary antibodies used for immunofluorescence (IF) and WB analyses were rabbit monoclonal anti-ALK (WB, 1:500–1000; IF, 1:200; #3333S, Cell Signaling Technology, Danvers, MA, USA), anti-pALK Y1507 (WB, 1:1000; #14678, Cell Signaling Technology) and anti-c-Met (WB, 1:1000; #8198, Cell Signaling Technology); rat monoclonal anti-α-tubulin (WB, 1:1000; IF, 1:800; MCA78G, Bio-Rad Laboratories, Hercules, CA, USA) and anti-HA (WB, 1:1000; IF, 1:400; 3F10, Roche, Basel, Switzerland); goat anti-Lamin B (WB, 1:200–500; sc-6216, Santa Cruz Biotechnology, Dallas, TX, USA); and mouse monoclonal anti-Actin (WB, 1:2000; A3853, MilliporeSigma, Burlington, MA, USA). Secondary antibodies included Alexa Fluor 555-labeled goat anti-rat IgG (1:1000; Life Technologies, Waltham, MA, USA); 488-labeled donkey anti-rabbit and anti-rat IgG (1:800; Life Technologies) for IF. The antibodies used for WB included horseradish peroxidase (HRP)-conjugated donkey anti-rabbit IgG (1:4000; 711-035-152), donkey anti-rat IgG (1:4000; 712-035-153), donkey anti-mouse IgG (1:4000; 715-305-151) from Jackson Immuno Research (West Grove, PA, USA), and bovine anti-goat (1:4000; sc-2350) from Santa Cruz Biotechnology. 

### 4.4. Transfection

Twenty pico-moles of small interfering RNA (siRNA) were transfected to HeLa S3 and SH-SY5Y cells using Lipofectamine 2000 (Invitrogen, Carlsbad, CA, USA). siMet #1 (5’-CACCUUUG AUAUAACUGUUTT-3’, Hs_MET_9694_s), siMet #2 (5’-CGGAUAUCAGCGAUCUUCUTT-3’, Hs_MET_9695_s), siMAD2 (5’-GUUGGAAGUUUCUUGUUCATT-3’, Hs01_00042213), siALK #1 (5’-GUCAUUACGAGGAUACCAUTT-3’, Hs_ALK_6329_s), and siALK #2 (5’-GAAGUGAAUAUUAAGCAUUTT-3’, Hs_ALK_6330_s) were purchased from Millipore Sigma (Burlington, MA, USA). siALK #3 targeted to the 3’-UTR region of ALK (5’-GUGAUAAAUACAAGGCCCATT-3’) and siMet #3 (5’-AAGCCAAUUUAUCAGGAGGTT-3’) were synthesized by MilliporeSigma. 

### 4.5. Cell Cycle Synchronization

For synchronization, SH-SY5Y and HeLa S3 cells were treated with 4 and 8 µM RO-3306 (Selleck Chemicals), respectively, for 20 h to arrest cells at the G2/M border. Cells were then released into fresh pre-warmed medium after washing with pre-warmed PBS containing Ca^2+^ and Mg^2+^ (PBS (+)) four times.

### 4.6. Western Blotting

Western blotting was performed as described previously [52]. Briefly, cells were lysed in SDS sample buffer containing 2 µg/mL aprotinin (Seikagaku Kogyo, Tokyo, Japan), 0.8 µg/mL pepstain A (Wako Pure Chemicals, Osaka, Japan), 2 µg/mL leupeptin (Nacalai Tesque), 2 mM PMSF (Nacalai Tesque), 20 mM β-glycerophosphate (MilliporeSigma), 50 mM NaF (Wako), and 10 mM Na_3_VO_4_ (Wako) and denatured at 40 and 100 °C for 20 and 5 min, respectively. Cell lysate components were separated by SDS-PAGE and transferred onto polyvinylidene difluoride membranes (Pall Corporation, Port Washington, NY, USA). Membranes were blocked with Blocking One (Nacalai Tesque) and incubated for 1 h at room temperature or overnight at 4 °C with primary and secondary antibodies diluted in Tris-buffered saline containing 5% Blocking One and 0.1% Tween20. Sequential reprobing of the membranes with various antibodies was performed after the inactivation of HRP by 0.1% NaN_3_. Proteins were detected with Chemi-Lumi One L (07880-70, Nacalai Tesque) and Clarity (#1705061, Bio-Rad, Hercules, CA, USA) using an image analyzer ChemiDoc XRSplus (Bio-Rad). Full-length blots of Figure 1, Figure 3, Figure 6, and Appendix A are shown in Appendix A.

### 4.7. Immunofluorescence Microscopy

Immunofluorescence staining was performed as described previously [52,53]. Cells were grown on coverslips and fixed with 4% formaldehyde in PBS for 20 min at room temperature. The fixed cells were permeabilized and blocked with PBS containing 0.1% saponin and 3% bovine serum albumin for 30 min and incubated with primary and secondary antibodies for 1 h each. For DNA staining, cells were treated with 1 µM Hoechst 33342 for 1 h together with secondary antibodies. The optical system included a U-FUNA filter cube (360–370 nm excitation, 420–460 nm emission), a U-FBNA filter cube (470–495 nm excitation, 510–550 nm emission), and a U-FRFP filter cube (535–555 nm excitation, 570–625 nm emission) for observing Hoechst 33342, Alexa Fluor 488, and Alexa Fluor 555 fluorescence, respectively. Captured images were edited using ImageJ (National Institutes of Health, Bethesda, MD, USA), Photoshop CC, and Illustrator CC software (Adobe, San Jose, CA, USA).

### 4.8. Time-Lapse Imaging

SH-SY5Y cells were seeded in 24-well plates and transfected with siCtrl and siALK #1 using Lipofectamine 2000. After 52 h, cells were synchronized with 4 µM RO-3306 for 20 h. After being washed with pre-warmed PBS (+) four times at 37°C on a water bath, pre-warmed DMEM containing 5% FBS and 0.1 µM Hoechst 33342 was added to the cultures. Immediately, the 24-well plate was set in the live cell chamber of an Operetta imaging system (PerkinElmer, Waltham, MA, USA), and live cell images of bright field and fluorescence of Hoechst 33342 were acquired every 3 min for 5 h at 37 °C in 5% CO_2_ [54]. Likewise, SH-SY5Y cells were treated with 1 µM crizotinib or DMSO immediately after RO-3306 release and live cell images were obtained as mentioned above. Similarly, H2228 cells were imaged in the presence of DMSO or 0.7 µM TAE684 using the same Operetta imaging system. Time-lapse images were captured every 5 min for 12 h. The duration of prophase/prometaphase was measured from mitotic entry (chromosomes condensation/cell round up) to metaphase (chromosomes alignment at the cell equator). The time from chromosomes alignment to their separation and from chromosomes separation to cleavage furrow completion were defined as metaphase and anaphase/telophase, respectively.

### 4.9. Proliferation Assay

Cell proliferation was assessed using a Cell Counting Kit-8 (Dojindo, Kumamoto, Japan) according to the manufacturer’s instructions as described previously [55]. SH-SY5Y cells (1 × 10^3^ per well) were seeded in 96-well plates, and the next day were cultured with crizotinib (0.01, 0.1, 1, and 10 µM), ceritinib (0.001, 0.01, 0.1, and 1 µM), and TAE684 (0.0001, 0.001, 0.01, 0.1, and 1 µM) for 2 days. Similarly, H2228 cells were cultured with crizotinib, ceritinib, and TAE684 (0.001, 0.01, 0.1, 1, and 10 µM) for 3 days. As a solvent control, cells were cultured in the presence of 0.1% DMSO. Based on absorbance at 450 nm by reduced 2-(2-methoxy-4-nitrophenyl)-3-(4-nitrophenyl)-5-(2,4-disulfophenyl)-2*H*-tetrazolium monosodium salt (WST-8), the number of cells was estimated. 

### 4.10. Plasmid

R777-E008 Hs.ALK-nostop (a gift from Dominic Esposito, plasmid 70292; Addgene, Watertown, MA, USA) was recombined with pLIX_402 (a gift from David Root, plasmid 41394; Addgene) [56] lentiviral plasmids using the Gateway LR reaction according to the manufacturer’s instructions (Invitrogen, CA, USA). Consequently, ALK was tagged with hemagglutinin (HA) at the C-terminus (ALK-HA) in the pLIX_402 vector. The lentiviral packaging plasmids pCAG-HIVgp and pCMV-VSV-G-RSV-Rev were a gift from Dr. Hiroyuki Miyoshi (Rikagau Kenkyusho Foundation (RIKEN) BioResource Center, Ibaraki, Japan)

### 4.11. Establishment of Stable Cell Lines by Lentiviral Transduction

To establish stable cell line capable of inducible overexpression of wild-type ALK, Lenti-X 293T cells were cotransfected with 1.2 µg of pLIX_402 vector harboring the ALK-HA construct, 0.8 µg of pCAG-HIVgp, and 0.8 µg of pCMV-VSV-G-RSV-Rev using PEI Max (Polysciences, Warrington, PA, USA) in a 35 mm dish. At 48 h after transfection, virus-containing media was harvested and passed through a 0.45 µm filter. A Mag4C LV magnetic kit (OZ Biosciences, San Diego, CA, USA) was used to concentrate viruses according to manufacturer’s instructions. SH-SY5Y cells were infected with the virus with 8 µg/mL polybrene (MilliporeSigma), and infected cells were selected with 1.5 µg/mL puromycin (StressMarq Biosciences, Victoria, BC, Canada).

### 4.12. Statistics

Statistical significance was determined using the Statcel add-in program for Microsoft Excel (OMS Publishing, Tokorozawa, Japan) with results from more than three independent experiments. The Bartlett test was used to determine the homogeneity of variance. For analysis among groups with equal variance, data were analyzed by one-way ANOVA followed by the Tukey–Kramer multiple comparisons test. The Student’s t-test was also performed to compare groups. A *p* value less than 5% was considered to be statistically significant. Statistical outliers were not excluded from the analysis. 

## 5. Conclusions

Inhibitors of ALK affect M phase progression in cells expressing wild-type ALK or EML4-ALK fusion via prolongation of the onset of anaphase due to defects in chromosome alignment. During M phase delay, SAC is activated indirectly. These findings are consistent with the proposition that ALK is a novel regulator of M phase and that delay in M phase could be a mechanism underlying suppression of cell proliferation caused by ALK inhibition. 

## Figures and Tables

**Figure 1 cancers-12-01054-f001:**
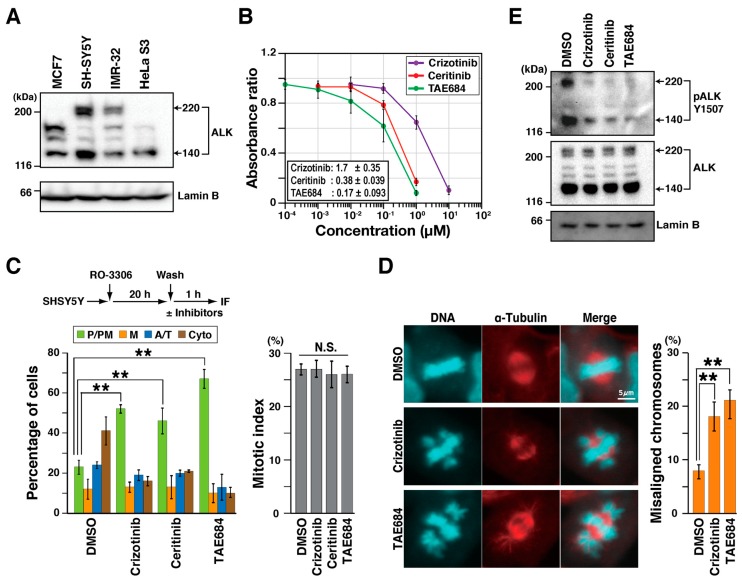
ALK inhibitors delay M phase progression. (**A**) Whole cell lysates were analyzed by western blotting with anti-ALK and anti-Lamin B (loading control) antibodies. (**B**) SH-SY5Y cells were treated with inhibitors for 2 days, and viable cells were determined by WST-8 assay. Relative values of absorbance to solvent control (dimethyl sulfoxide, DMSO) are shown as mean ± SD, calculated from three independent experiments. (**C**) SH-SY5Y cells were synchronized with 4 µM RO-3306 for 20 h. After release, cells were treated with 1 µM crizotinib, 0.5 µM ceritinib, and 0.3 µM TAE684 for 1 h and stained for α-tubulin and DNA. Based on microtubule and chromosome morphology, M phase cells were classified into four groups: prophase/prometaphase (P/PM), metaphase (M), anaphase/telophase (A/T), and cytokinesis (Cyto). The percentages of cells of each group (n > 246 in each experiment) and associated mitotic indices (*n* > 453 in each experiment) are plotted as the mean ± SD, calculated from three independent experiments. (**D**) The number of cells with misaligned chromosomes was counted under a microscope, and their percentages are plotted as the mean ± SD of three independent experiments (*n* > 246 in each experiment). Representative images of cells with misaligned chromosomes are shown; scale bar = 5 µm. The Tukey–Kramer multiple comparison test was used to calculate *p* values. ** *p* < 0.01; N.S., not significant. (**E**) SH-SY5Y cells were treated with inhibitors in the same concentrations as in (**C**) for 1 h, and whole cell lysates were analyzed by Western blotting using anti-pALK-Y1507 and anti-ALK antibodies. Lamin B was incorporated as a loading control.

**Figure 2 cancers-12-01054-f002:**
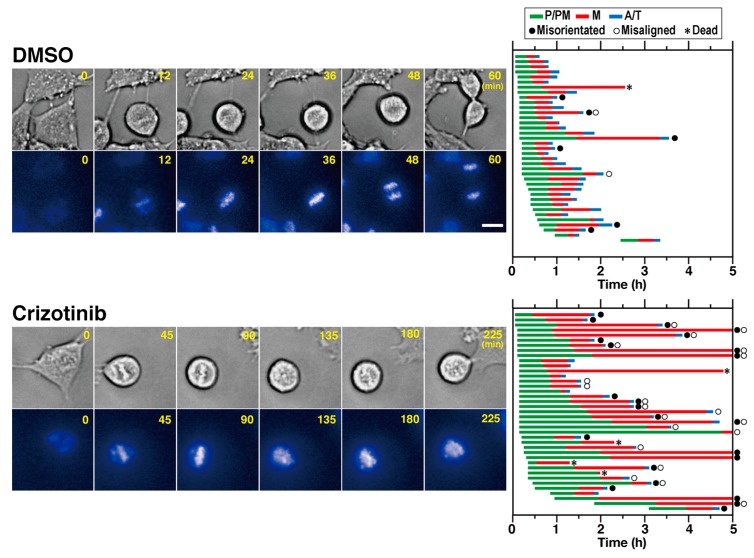
Crizotinib delays both prophase/prometaphase and metaphase. The SH-SY5Y cells were synchronized with 4 µM RO-3306 and their M-phase progression in presence and absence of 1 µM crizotinib were observed every 3 min for 5 h using time-lapse imaging. Representative images of solvent control (DMSO) and crizotinib-treated cells are shown. Scale bar = 10 µm. Durations of prophase and prometaphase (P/PM, green), metaphase (M, red), and anaphase and telophase (A/T, blue) for 37 control and 39 crizotinib-treated cells are shown in the graph. Cells showing misoriented spindle, misaligned chromosomes, and cell death are indicated by closed circle, open circle, and asterisk, respectively.

**Figure 3 cancers-12-01054-f003:**
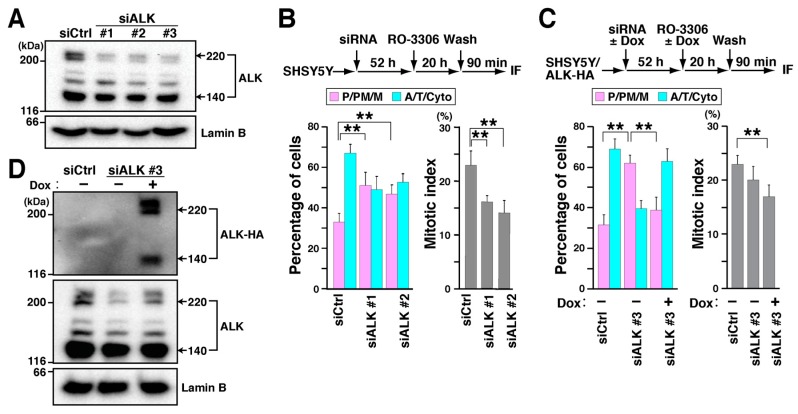
ALK is responsible for ALK-inhibitor-induced M phase delay. (**A**) SH-SY5Y cells were transfected with ALK-targeting siRNAs (siALK #1, #2, and #3) or nontargeting siCtrl. At 72 h after transfection, whole cell lysates were prepared and analyzed by western blotting. (**B**) At 52 h after siRNA transfection, SH-SY5Y cells were synchronized with 4 µM RO-3306 and then cultured in fresh media for 90 min. Based on microtubule and chromosome morphology, M phase cells were classified into two groups: before (P/PM/M) or after (A/T/Cyto) anaphase onset. Percentages of cells (*n* > 243 in each experiment) and mitotic indices (*n* > 503 in each experiment) are plotted as the mean ± SD calculated from three independent experiments. (**C**,**D**) SH-SY5Y/ALK-HA cells were transfected with siRNA targeting the 3’-UTR region of ALK (siALK #3) or nontargeting siCtrl and were simultaneously treated with 5 µg/mL Dox. (**C**) Fifty-two hours after transfection, the cells were synchronized with RO-3306 and M phase progression was analyzed as described in (**B**). Percentages of cells (*n* > 193 in each experiment) and mitotic indices (n > 413 in each experiment) are plotted as the mean ± SD of three independent experiments. The Tukey–Kramer multiple comparison test was used to calculate *p* values. ** *p* < 0.01. (**D**) At 72 h after transfection, whole cell lysates were analyzed by Western blotting with anti-HA, anti-ALK, and anti-lamin B antibodies.

**Figure 4 cancers-12-01054-f004:**
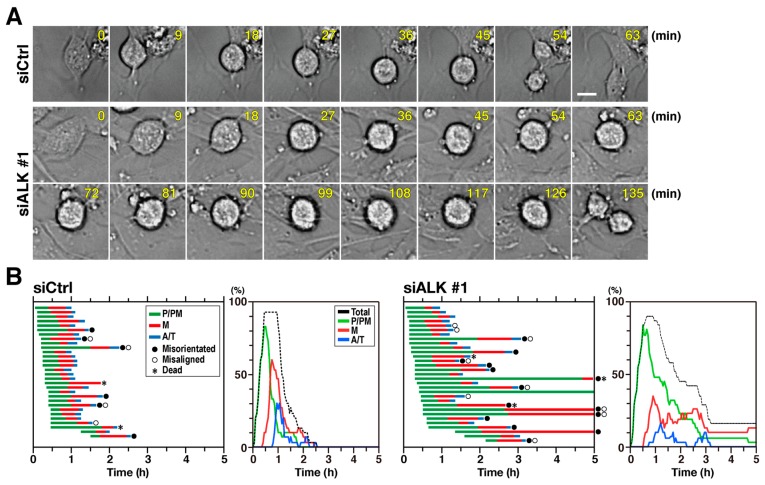
ALK inhibition delays both prophase/prometaphase and metaphase. SH-SY5Y cells were transfected with ALK-targeting siRNA (siALK #1) or nontargeting siCtrl. At 52 h after siRNA transfection, cells were synchronized with 4 µM RO-3306 and mitotic progression was monitored every 3 min for 5 h by time-lapse imaging. (**A**) Representative images of siCtrl-transfected cells and ALK knockdown cells. Scale bar = 10 µm. (**B**) Duration of prophase and prometaphase (P/PM, green), metaphase (M, red), and anaphase and telophase (A/T, blue) for 30 individual control cells and 31 individual siALK–transfected cells, was measured as described in material and methods. Line graphs show percentages of total mitotic cells (black), cells in P/PM (green), M (red), and A/T (blue) plotted against time.

**Figure 5 cancers-12-01054-f005:**
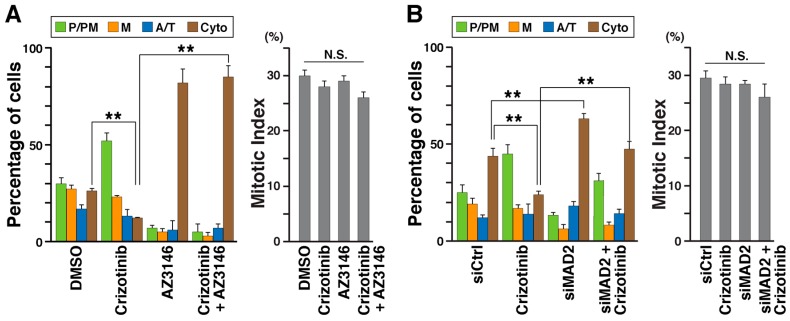
SAC is involved in crizotinib-induced M phase delay. (**A**) SH-SY5Y cells were synchronized with 4 µM RO-3306 and released in the presence of 0.5 µM crizotinib, 4 µM AZ3146, or a combination of both for 45 min. (**B**) SH-SY5Y cells were transfected with siMAD2 and 28 h later the cells were synchronized using RO-3306. Then, they were released with or without 0.5 µM crizotinib treatment for 45 min. M phase cells were classified into four groups as described in Figure 1C. The percentages of cells in subphases (**A**, *n* > 256; **B**, n > 240, in each experiment) and mitotic indices (**A**, *n* > 464; **B**, *n* > 470, in each experiment) are plotted as mean ± SD, calculated from three independent experiments. The Tukey–Kramer multiple comparison test was used to calculate *p* values. ** *p* < 0.01; N.S., not significant.

**Figure 6 cancers-12-01054-f006:**
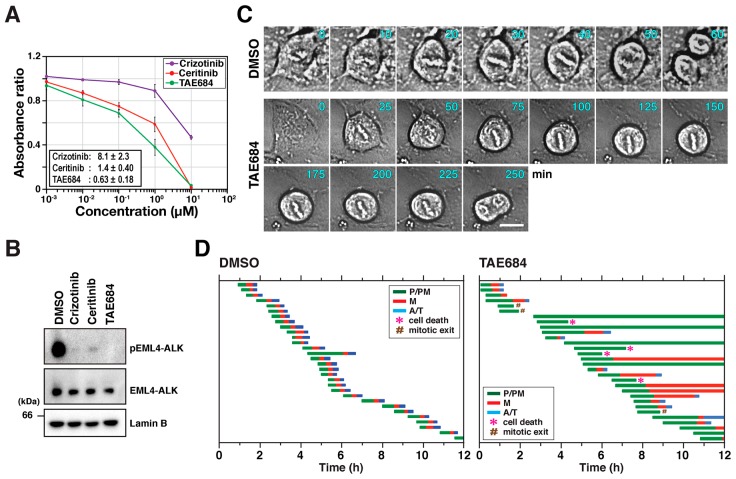
Inhibition of EML4-ALK fusion delays M phase progression. (**A**) H2228 cells were treated with inhibitors for 3 days, and viable cells were determined by WST-8 assay. Relative values of absorbance to solvent control (DMSO) are shown as mean ± SD calculated from three independent experiments. (**B**) H2228 cells were treated with DMSO, 8.2 µM crizotinib, 1.4 µM ceritinib, and 0.7 µM TAE684 for 1 h. Whole cell lysates were analyzed by WB with indicated antibodies. (**C**,**D**) Time-lapse imaging of asynchronous H2228 cells was performed for 12 h in the presence of DMSO and 0.7 µM TAE684. (**C**) Representative images of solvent control (DMSO) and TAE684-treated cells; scale bar = 10 µm. (**D**) Duration of P/PM (green), M (red), and A/T (blue) in individual cells (DMSO and TAE684, *n* = 30). *, cell death; #, mitotic exit.

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
