# Peer review of "ALK Inhibitors-Induced M Phase Delay Contributes to the Suppression of Cell Proliferation"

_cancers, 2020, doi:10.3390/cancers12041054_

Round 1

Reviewer 1 Report

The authors have satisfactorily addressed most of my comments and made the necessary changes to the manuscript. 

This manuscript is a resubmission of an earlier submission. The following is a list of the peer review reports and author responses from that submission.

Round 1

Reviewer 1 Report

The study is flawless and indicates that aurora kinase inhibitors and/or anti-microtubule drugs and PLK1 inhibitors can enhance the effect of ALK inhibitors in ALK fusion NSCLC  and neuroblastoma patients with ALK mutations.

A few points for the authors:

1. The PFS and overall survival with ALK inhibitors in ALK NSCLC patients is much better that what is inferred in the manuscript. (Peters et al "Alectinib vs crizotinib...." NEJM 2017). There are other potent inhibitors, like brigatinib.

2. The authors use the EML-4ALK H2228 cells, but there is a more common cell line, H3122 cells. 

3. The mechanisms of resistance have been studied in depth and perhaps a summary of the mechanisms should be included in the manuscript. (Tanizaki et al BJC 2012; Yamada et al CCR 2012; Hrustanovic et al Nat Med 2015; Dardaei et al Nat Med 2018, Katayama et al Science Trans Med 2012, among others)

Reviewer 2 Report

In the manuscript "ALK inhibitors-induced M phase delay contributes to the suppression of cell proliferation" Munira et al. evaluated the effects of ALK inhibitors on M phase progression. The authors showed that ALK inhibitors, as well as ALK silencing, suppressed proliferation and caused misalignment of chromosomes in the neuroblastoma cells SHSY5Y and EML4-ALK-positive lung carcinoma cells H2228. Time-lapse imaging experiments revealed a delayed onset of anaphase. Finally, the authors showed that the combination of the Mps1 inhibitor, AZ3146, with crizotinib led to a release from inhibitor-induced M phase delay, suggesting that spindle assembly checkpoint may be activated in ALK-inhibited cells. The authors speculate that ALK plays a role in M phase regulation and ALK inhibition may contribute to the suppression of cell proliferation in ALK-expressing cancer cells.

The study is well written and concise, however, its results are mainly descriptive and several key points are not experimentally addressed. Importantly, the molecular mechanisms linking ALK inhibition with mitotic arrest are not identified, and whether defects in spindle formation are directly or indirectly activated by ALK inhibition remains an open question.

Major points:

  1. Figure 1A and Line 275-285: Several bands detected by ALK antibodies, could be non-specific signals, as they are detected also in ALK-negative cell lines. Figure 1A should be supported by appropriate negative controls.
  2. Figure 1C: It is shown that 1h treatment induces a dramatic mitotic arrest.  Demonstrate that cells are alive and efficiently synchronized by cell cycle FACS analysis.
  3. In Figure 2B is not clear whether the increased number of cells arrested in P/PM/M phase and the reduced mitotic index is a consequence of increased cell death induced by ALK inhibition.
  4. In figure 2D endogenous ALK expression should be visible in siCtrl cells as in Figure 2A.
  5. Figure 3: Time-lapse imaging describes that ALK silencing in SHSY5Y cells prolongs P/PM and M. It would be interesting to know whether the delay of M phase progression is followed by cell death or aberrant cytokinesis (as described in Figure 5D).
  6. Figure 4: The authors observed that the treatment of SHSY5Y cells with the MPS1 inhibitor was able to release the M-phase delay induced by ALK inactivation. The molecular mechanisms linking ALK inhibition with mitotic arrest are not showed, as well as the molecular and biochemical information on the status of mitotic checkpoint and anaphase-promoting complexes (MCC and APC/C) are missing.
  7. Discuss whether the increased chromosome misalignment induced by ALK inhibitors could contribute to genomic instability, clonal evolution, and drug resistance.

Minor points:

Introduction: Anaplastic Large Cell Lymphoma should be cited as the disease in which ALK translocations have been first identified.